# The Immune Landscape of Osteosarcoma: Implications for Prognosis and Treatment Response

**DOI:** 10.3390/cells10071668

**Published:** 2021-07-02

**Authors:** Caterina Cascini, Claudia Chiodoni

**Affiliations:** Molecular Immunology Unit, Department of Research, Fondazione IRCCS Istituto Nazionale dei Tumori, 20133 Milan, Italy; caterina.cascini@istitutotumori.mi.it

**Keywords:** osteosarcoma, tumor microenvironment, macrophages, bone marrow, osteoclasts, mesenchymal stem cells

## Abstract

Osteosarcoma (OS) is a high-grade malignant stromal tumor composed of mesenchymal cells producing osteoid and immature bone, with a peak of incidence in the second decade of life. Hence, although relatively rare, the social impact of this neoplasm is particularly relevant. Differently from carcinomas, molecular genetics and the role of the tumor microenvironment in the development and progression of OS are mainly unknown. Indeed, while the tumor microenvironment has been widely studied in other solid tumor types and its contribution to tumor progression has been definitely established, tumor–stroma interaction in OS has been quite neglected for years. Only recently have new insights been gained, also thanks to the availability of new technologies and bioinformatics tools. A better understanding of the cross-talk between the bone microenvironment, including immune and stromal cells, and OS will be key not only for a deeper knowledge of osteosarcoma pathophysiology, but also for the development of novel therapeutic strategies. In this review, we summarize the current knowledge about the tumor microenvironment in OS, mainly focusing on immune cells, discussing their role and implication for disease prognosis and treatment response.

## 1. Introduction

Osteosarcoma (OS), the most frequent bone tumor in children and adolescents, is a high-grade malignancy characterized by the formation of an osteoid matrix and immature bone, mainly in the metaphysis and diaphysis of long bones [1]. OS shows a very aggressive behavior with lung micro-metastases often already present at the time of diagnosis. Although primary lesions can be efficiently treated with surgical resection and (neo)adjuvant chemotherapy administration (doxorubicin, cisplatin, methotrexate, and ifosfamide), no effective treatment options are yet available for metastatic OS [2]. Indeed, whereas the five-year survival rate is more than 78% for localized disease, it drops to 25% for metastatic or relapsing OS [3].

Although the precise cell of origin for OS is still unclear, some evidence indicates that it could arise from mesenchymal stem cells (MSCs) or from osteoblastic progenitors unable to proceed to terminal differentiation [4]. In the putative cell of origin, many genetic alterations, such as copy number variants and multiple fusion sequences in chromosomes, occur with high frequency and draw a complex genetic scenario that hinders the identification of a unique driver mutation for further therapeutic strategies [5]. OS aggressiveness and poor prognosis for metastatic patients, together with the absence of targeted therapies, make it urgent to identify new therapeutic approaches to improve the overall survival rate of OS patients.

In the search for new therapeutic targets, in recent decades, the tumor microenvironment (TME) has gained increasing attention in several tumor types. However, differently from most carcinomas, a clear picture of the TME and of its role in the development and progression of OS is still lacking. Indeed, while it has been widely studied in other solid tumor types and its contribution to tumor progression has definitely been established [6,7], tumor–stroma interaction in OS has been quite neglected thus far and the few available findings are sometimes contradictory with other tumor types. In addition, bone marrow (BM) is a very specialized microenvironment, containing highly heterogeneous cell types, ranging from hematopoietic cells to mesenchymal, vascular, and neuronal cells. However, in the last few years, also thanks to recent technological advances, new insights in the OS TME have been gained. This review tries to summarize the state of the art of the TME in OS, with a particular focus on tumor-infiltrating immune cells and their correlation with patients’ prognosis and response to treatment.

## 2. The Heterogenous Bone Marrow Microenvironment

Differently from other primary tumors, the tumor microenvironment of OS is the BM, a highly dynamic environment composed of bone cells, immune cells, and stromal and vascular cells, embedded in a mineralized extracellular matrix. In normal physiological conditions, all these cell types act in coordination to maintain bone and hematopoietic homeostasis. The cross-talk between these cell types is tightly regulated by cytokines and growth factors, which are also responsible for bone remodeling through the balance between the activities of osteoclasts and osteoblasts. The concept that the immune cells and bone tissue are strictly interconnected has recently led to the development of a new interdisciplinary field, “osteoimmunology” [8]. While the original idea was mainly focused on the role of immune cells in bone damage in pathological conditions, such as inflammatory diseases or neoplastic malignancies, accumulating evidence has demonstrated that bone cells can also reciprocally regulate immune cells and hematopoiesis. In this context, it fits well the idea that the pathophysiology of OS is strictly dependent not only on the molecular events underlining osteoblast differentiation, but also on the interaction with the other cell types residing in the BM [9] (Figure 1).

### 2.1. Immune Cells

Immune cell types come in different “flavors,” depending on the specific context, local microenvironment, and timing. This is true for most immune cells, belonging either to the myeloid or lymphoid compartments. For example, macrophages are generally dichotomized into M1 inflammatory/anti-tumor and M2 pro-tumoral/pro-angiogenic macrophages, and CD4 T cells can be Th1, Th2, Th17, T regulatory cells, or other newly identified CD4 T cell subsets that may have either pro- or anti-tumor characteristics [10,11]. However, a plethora of recent evidence clearly indicates that most of these subsets are not completely distinct from one another, but rather they are a plastic continuum that can shift into one or the other depending on the stimuli they receive from the microenvironment they reside in [12,13].

#### 2.1.1. Macrophages

Macrophages are a heterogenous cell population often representing one of the most abundant immune cell subsets in the TME. As highly plastic cells, tumor-associated macrophages (TAMs) are able to adapt to various signals derived from a specific TME, thus developing different polarization status. Nonetheless, TAMs generally show the so-called M2 pro-tumoral phenotype and their association with tumor aggressiveness and poor prognosis has been well established for several type of cancers [14]. Differently, in the context of OS, the role of TAMs is not clearly defined yet, with pieces of evidence suggesting a pro-tumoral role, as for most tumor types, and a few others indicating a correlation with good prognosis, indicative of anti-tumor activity. Along the latter line, Buddingh and colleagues, carrying out gene expression profiling (GEP) on pre-chemotherapy biopsies of patients who did and did not develop metastases within five years, found an up-regulation of macrophage-related genes in non-metastatic patients [15]. Additionally, by immunohistochemistry (IHC), they showed that TAMs are associated with significantly better overall survival independently from their M1 (CD14/HLA-DRα-positive) or M2 phenotypic TAMs (CD14/CD163-positive). Similarly, Gomez-Brouchet’s group showed that high levels of CD163^+^ macrophages are associated with a better prognosis and, interestingly, with a prolonged metastasis progression-free survival [16]. In contrast, Han and colleagues described that CD68 expression is significantly higher in OS samples from patients with metastasis [17]. The expression of CD68 also correlates with that of COX-2, and they also found an increased up-regulation of COX-2 in lung metastases compared to the primary OS lesion of the same patients. Unexpectedly, CD163, which is considered a biomarker of M2 macrophages, did not show a significant difference. In a different study, the same group, in order to study the regulatory effect of the TME on T cell function in OS, analyzed the peripheral blood (PB) and tumor-infiltrating T cells (TILs) in patients with localized OS disease and found that the frequency of CD163^+^ TAMs is directly correlated with that of suppressed TIM-3^+^ PD-1^+^ T cells in tumor lesions [18].

In agreement with the idea that M1 TAMs may have anti-tumor activity, Dumars et al. revealed a higher infiltration of the INOS^+^ M1 subtype in the OS tumors of non-metastatic patients [19].

Besides the analysis of patients’ samples, either through transcriptomic profiling or IHC, and the definition of their correlation with prognosis, pre-clinical studies have also investigated the role of TAMs in OS development and aggressiveness. Using a mouse model of human OS xenograft, Xiao et al. showed that the macrophages recruited in the tumor show an M2 subtype and that their deletion with liposome-encapsulated clodronate decreases tumor growth [20]. Along the same line, Shao et al. reported that M2 macrophages contribute to OS initiation and progression by orthotopic co-injection in the mouse tibia of the K7M2 OS cell line with RAW264.7 macrophagic cells [21]. Treatment with all-trans retinoic acid (ATRA) delayed OS growth by impacting on macrophage M2 polarization. The authors, searching for the mechanism responsible for their phenotype, showed that ATRA suppresses the colony formation and sarcosphere formation of OS cells by decreasing M2 polarization [21].

Han’s group, in the same paper mentioned above, also provided some functional evidence for a pro-tumoral role of TAM by co-culture experiments with human OS cell lines. M0 and M2 macrophages derived from the THP-1 human monocytic cell line promoted the migration and invasion of OS cell lines more significantly than M1 macrophages. Interestingly, macrophage-conditioned medium induced changes in the epithelial-to-mesenchymal transition (EMT) markers in OS cells toward a more mesenchymal phenotype, through the down-regulation of E-cadherin and up-regulation of N-cadherin, vimentin, and EMT transcription factors such as ZEB-1 and SNAIL [17]. Despite the fact that many of these pre-clinical works used established macrophagic cell lines and not fresh macrophages, they all point to a pro-tumoral role of M2 macrophages. Table 1 summarizes the most recent studies on the roles of macrophages in OS.

#### 2.1.2. Other Myeloid Cells

The so-called myeloid-derived suppressor cells (MDSCs) constitute another immune cell subset that has been shown to play a key role in favoring tumor progression in different tumor types. MDSCs comprise a heterogeneous population of mostly immature myeloid cells that are characterized by a pathological state of activation and immune suppressive activity. Two major subsets of MDSCs have been identified: Monocytic (M-MDSCs) and polymorphonuclear (PMN-MDSCs), being the first one similar to monocytes and characterized by high plasticity, and the second sharing phenotypic and morphologic features with neutrophils [26,27]. 

As for other tumor types, few reports point to a role of MDSCs in tumor progression and aggressiveness also in OS. Uehara and colleagues suggest a role for MDSCs in OS progression by using an in vivo OS mouse model (K7M2 cell line) treated with metformin, a type 2 diabetes drug. They found that metformin treatment is able to reduce tumor growth, which was associated with a significant decrease in tumor-infiltrating PMN-MDSCs (whereas no major difference was detected for M-MDSCs). By using an antibody to deplete CD11b^+^ cells in combination with metformin, the growth inhibitory effect of metformin was abrogated, suggesting that its activity is exerted through myeloid cells and that this population contributes to OS growth [28]. 

In a recent work, Jiang et al. showed that OS tissues are heavily infiltrated by CD11b^+^ MDSCs that also express the CXCR4^+^ marker. On the contrary, CD8 T cells are largely absent in CD11b-infiltrated tumors [29]. To functionally prove the relevance of their findings, they moved to an in vivo OS mouse model and tested the efficacy of AMD3100, a CXCR4 inhibitor, either alone or in combination with an anti PD-1 antibody. AMD3100 alone was already able to slightly inhibit OS tumor growth, and the combination exerted the highest inhibitory effect. The analysis of tumor-infiltrating immune cells showed that the most relevant population is indeed represented by CXCR4-expressing MDSCs and that treatment with AMD3100 significantly reduces this population. With additional in vitro experiments, they demonstrated that the CXCL12/CXCR4 axis likely promotes the migration and survival of MDSCs via the activation of the PI3K/Akt pathway in OS cells [29].

#### 2.1.3. T Lymphocytes and Other Immune Cell Types

The mechanisms underlying the interplay between T lymphocytes and tumor cells in the TME are intricate and still not completely understood. Despite the fact that a high percentage of T cells may infiltrate a tumor mass, cancer cells are usually able to evade immune attack through immunoediting or other escape mechanisms [30,31,32]. Among these mechanisms, the immune checkpoints, such as PD-1/PD-L1, which play a crucial role in maintaining self-tolerance and in tuning the duration and amplitude of physiological immune responses, are hijacked by cancer cells to evade immune attacks. Immune checkpoint inhibitors have therefore been designed to target such negative regulatory pathways of T cell function, and they have indeed been shown to achieve considerable clinical results, at least for specific tumor types, such as melanoma and lung cancer [33,34]. Differently, in sarcomas, immune checkpoint inhibitors have not shown compelling clinical results thus far, and the expression and role of PD-1/PD-L1 axis are not yet clearly defined [35].

In the context of OS, Shen and colleagues analyzed PD-L1 expression in 38 OS samples by RT-PCR and IHC on tissue microarrays [36]. Besides detecting a very high variability in PD-L1 expression among samples, they found that the median overall survival for PD-L1-low patients is 89 months compared to 28 months for PD-L1-high patients (*p* = 0.054). Additionally, they reported a positive correlation between PD-L1 mRNA expression and TILs.

Similarly, the group of Koirala evaluated, in two OS patient cohorts, the expression of PD-L1 and the presence of TILs and antigen-presenting cells (APCs) by IHC, and correlated PD-L1 expression with immune cell infiltration and event-free survival [37]. Only a small fraction of OS samples (between 7% and 25%) was shown to express PD-L1 and the positive samples demonstrated <25% PD-L1 staining, quantified as the percentage of positive tumor cells compared to the tumor volume. Tumors expressing PD-L1, in comparison to those that were PD-L1 negative, were more infiltrated by CD3^+^ T cells, CD56^+^ NK cells, CD68^+^ macrophages, and CD1a^+^ dendritic cells. In this study, PD-L1 expression was associated with significantly poorer five-year event-free survival, thus confirming the results from Shen’s group.

Recently, Toda’s group evaluated the expression of PD-L1, IDO1, CD3, CD4, and CD8 in tumor tissues collected by biopsy or surgical resection from 56 OS patients, by comparing primary and metastatic lesions, and samples before neo-adjuvant chemotherapy (NAC) and after NAC [38]. In pre-NAC primary lesions, only 17% showed PD-L1 expression and 12% IDO1 expression, and the two molecules were co-expressed in most cases. In post-NAC metastatic samples, the frequency of PD-L1 and IDO1 expression was increased, but it was not associated with poor prognosis. PD-L1 expression was significantly associated with infiltration of CD3, CD4, and CD8 T cells, in line with previous studies, while IDO1 positivity with CD3 and CD4 T cells [38].

Palmerini and colleagues, hypothesizing that the immune infiltrate correlated with superior survival, analyzed 129 primary OS in tissue microarrays by IHC to detect the different immune cell types [39]. They found that 90% of samples were CD3-positive, with 86% also showing a marked CD8 positivity. In agreement with the study of Koirala and colleagues, they reported the expression of PD-L1, as well as of PD-1 on TILs, in only 22% of samples. Univariate analysis showed better five-year overall survival for patients with CD8 T cells positive for Tia1, a marker associated with cytotoxic T cells, but not for CD8 T cells overall, in contrast to Koirala’s study. No statistically significant difference was observed in five-year overall survival for PD-1, FOXP3, CD68, CD20, Arginase-1, CD303, or CD163 expression. Interestingly, a recent multicentric retrospective study demonstrated that a high ratio of CD8^+^/FOXP3^+^ T cells (above the median value of 3.08) in biopsy specimens taken before chemotherapy is indicative of prolonged survival [40].

Evidence of a potential role for T regulatory cells (Tregs) in OS progression comes also from recent preclinical studies. A study from the group of Yoshida reported that the anti-tumor efficacy of the anti-PD-1 antibody in an LM8 OS mouse model is likely mediated by the reduction of the number of FOXP3^+^ Tregs and an increase in TILs in the TME [41]. Using the same OS mouse model, Takenada and colleagues showed that the combination of high-dose local radiation and anti-CTLA4 antibody administration is able to halt tumor growth not only of the treated lesion, but also of a contralateral, untreated tumor (abscopal effect). Also in this work, treatment was able to severely reduce the number of FOXP3^+^ Tregs in both treated and untreated lesion [42]. Another T cell sub-population that has been implicated in tumor progression, although with controversial data, is the T helper 17 (Th17) subset [43]. Although no direct evidence thus far indicates a role for this subset in OS, the group of Honorati suggested that IL-17R in OS tumors might represent a marker for metastatic potential [44]. Controversial data are available also for γδ T cells. Due to their consistent representation in the TME and their cytotoxic ability, this cell subset has gained attention for immunotherapy-based approaches also in OS [45]. Several reports have indicated that γδ T cells are able to kill OS cells and that their activity can be increased by adjuvant treatment, such as with zoledronic acid (ZA) [46,47]. A recent work from Wang and colleagues reported that γδ T cells enhanced their cytotoxic activity against OS after the administration of valproic acid in combination with ZA, both in vitro and in vivo, through the accumulation of intermediates of the mevalonate pathway [48]. Similarly, Wang’s group demonstrated how the combination of γδ T cell therapy with decitabine, a DNA demetilating agent, sensitizes γδ T cells to recognize tumor antigens. Decitabine unveils NKG2DL expression in tumor cells that, in turn, binds the NKG2D receptor to γδ T cells and this interaction is able to activate a rapid cytotoxic response resulting in inhibition of tumor aggressive features in vitro and a reduction in tumor growth in vivo [49].

Despite these reports, accumulating evidence in different tumor types now points to γδ T cells as drivers of tumor progression through the establishment of an immune-suppressive environment, the induction of angiogenesis, and the inhibition of anti-tumor immunity, but no such data are available thus far in the context of OS [50].

### 2.2. Mesenchymal Stem Cells

Mesenchymal stem cells (MSCs) or an osteogenic lineage-committed progenitor are considered to be the potential cell of origin of OS. Besides this central role in OS transformation, among non-immune BM cells that have been shown to contribute to OS progression, MSCs represent a key subset. In the BM, MSCs actively regulate bone and hematopoietic homeostasis, being at the same time sensor and modulator, by receiving different signals from the local microenvironment, and in turn secreting growth factors, chemokines, and cytokines (for a comprehensive review, see [51]). Arising in the same bone microenvironment, OS cells are also affected by neighboring MSCs, but it is now clear that the interaction between MSCs and OS cells is bidirectional, with OS also influencing MSCs and reprogramming their phenotype to their advantage.

MSCs secrete a plethora of factors known to favor OS growth, metastatic dissemination, and angiogenesis, such as chemokines (CCL5 and CXCL12), cytokines (IL-6), and angiogenic factors (VEGF and PDGF) [52]. The effect of IL-6 and VEGF released by MSCs on OS growth has been demonstrated in several studies by co-culture in vitro experiments of MSCs and OS cells [53,54]. An elegant report on the bidirectional interaction between OS cells and MSCs was recently published by Pietrovito and colleagues, who showed that BM-derived MSCs migrate in vitro toward OS cells that release monocyte chemoattractant protein (MCP)-1, growth-regulated oncogene (GRO)-α, and transforming growth factor (TGF)-β1. Once in contact with OS cells, MSCs trans-differentiate into cancer-associated fibroblasts, which, in turn, release cytokines such as IL-6 and IL-8 in the tumor microenvironment, which promote OS cell motility, invasiveness, and trans-endothelial migration [55].

Besides the direct secretion of these factors, MSCs can also carry them in extracellular vesicle (EV) cargos, which are then released. MSC-secreted EVs may contain, in addition to different proteins, tumor-promoting microRNAs and metabolites, such as glutamate and lactate [56]. On the other side, OS cells also educate MSCs through their release of EV. For example, Baglio et al. reported that metastatic human OS cells release EVs containing a membrane-bound form of TGFβ, which are able to induce human MSCs to produce IL6 in vitro. Using a preclinical mouse model, they demonstrated that such tumor-educated MSCs promote OS growth and lung metastasis [57]. 

### 2.3. Osteoclasts

OS development is often associated with local bone osteolysis, which causes pain and bone fragility. As osteoclasts (OCs) are the cells responsible for bone resorption, it could be a logical assumption that OCs may be involved in OS development and progression. However, the role of OCs in the pathogenesis of OS is still debated. OS cells secrete factors, such as bone morphogenic proteins (BMPs), RANKL, and IL-6, which can stimulate osteoclastogenesis, which leads to increased osteolysis and to the consequent release of TGFβ, insulin-like growth factors (IGFs), fibroblast growth factors (FGFs), and others facilitating OS tumor growth from the bone matrix [58]. As an indirect indication that OC may contribute to OS tumor growth, Avnet et al. analyzed the presence of osteoclasts by IHC and through the expression of osteoclast-related gene mRNAs in tumor biopsies from a small cohort of OS patients. The authors found tumor-associated osteoclasts in 63% and 75% of cases, respectively, and the expression of tartrate-resistant acid phosphatase 5b (TRACP 5b), a marker of osteoclast presence, was significantly higher in metastatic patients compared to non-metastatic patients [59]. Interestingly, a very recent work from Zhou et al., based on single-cell RNAseq, showed that OCs were present not only in primary and recurrent OS tumors, but also in lung metastasis from OS patients, supporting the idea that OCs may have a role in OS cell growth and dissemination [22]. In agreement with these findings, a few studies have shown that OC-ablative therapies could reduce OS tumor growth and metastasis [60,61]. 

RANK-Ligand (RANK-L), a molecule produced by osteoblasts that increases OC activity, has indeed been proposed as a promising therapeutic target for the treatment of OS in which an imbalance of the RANK/RANK-L/OPG pathway often occurs. Along this line, Akiyama and colleagues showed, in an orthotopic model of OS, that the administration of RANK-Fc, a potent RANKL antagonist and inhibitor of OC formation and activity, is able to reduce lung metastasis, preserve bone structure, and reduce TRAP^+^ (tartrate-resistant acid phosphatase) OCs in OS-bearing bone [60,61]. More recently, Chen’s group, in a very elegant work, developed a series of genetically engineered mouse models of OS and used them to study the role of RANK-L in the disease, demonstrating that RANK-L blockade with RANK-Fc inhibits tumor progression and lung metastasis, improving survival. Additionally, early administration of RANK-Fc completely prevents tumorigenesis in mice highly predisposed to developing OS [62]. This work provided a rationale to consider RANK-L blockade for the treatment of RANK-L-overexpressing OS in humans, and indeed, a few clinical trials are ongoing to investigate the effect of Denosumab, an antibody against RANK-L, in the treatment of recurrent or refractory OS.

Among anti-resorptive agents, zoledronic acid (ZA), a third-generation bisphosphonate, effectively inhibits OC bone resorption and is widely utilized in the treatment of metastatic bone diseases for its acceptable safety profile and tolerability. In vitro and in vivo studies have suggested its use also in the treatment of OS (for a comprehensive review on the molecular and cellular mechanism of ZA in OS, refer to [63]). However, the few clinical studies on the efficacy of ZA for the treatment of OS have demonstrated conflicting results and larger clinical trials, as well as further molecular and in vivo pre-clinical studies are required to solve this issue [63].

As an example of the few studies indicating that OC-ablative therapies may be detrimental in the case of OS, Endo-Munoz et al. showed that the ablation of osteoclasts with ZA increases the number of lung metastases in an orthotopic OS model, whereas fulvestrant, an estrogen antagonist that blocks the estrogen receptor leading to increased numbers and activity of osteoclasts, increases osteoclast numbers and reduces metastatic lesions [64]. One hypothesis could be that OCs may either favor or inhibit OS growth depending on whether it is a primary or a secondary bone tumor. In the first case, OCs may contribute to the formation of a niche within the bone that nurtures the growth of OS cells in the early stages. Instead, in later stages of the disease, the inhibition of osteoclastogenesis and/or loss of OCs may favor tumor cell egress from the local tumor site and metastatic spread. Additionally, to make the situation even more complicated, it should be kept in mind that RANK-L-directed therapies (Denosumab and RANK-Fc) and bisphosphonates have different mechanisms of action, the first ones targeting osteoblasts and likely OS cells, whereas the second ones directly affecting OCs.

## 3. Definition of Osteosarcoma Immune Landscape by Transcriptomic Analysis and Bioinformatics Tools

The recent advances in transcriptomic profiling, from classical gene expression to RNA sequencing and single-cell analysis, and of very sophisticated bioinformatic tools able to provide an estimation of the abundance of specific cell types in a mixed cell population have certainly widened our knowledge of the TME in OS. CIBERSORT, a deconvolution method from the Alizadeh Lab developed by Newman et al. that infers cell composition of complex tissue from its GEP [65], represents one of the first available tools. The advantage of these tools is that they can be applied to publicly available gene expression datasets produced with completely different aims. In line with this idea, Niu and colleagues integrated different OS microarray datasets from the Gene Expression Omnibus (GEO) database to identify the differentially expressed genes (DEGs) between primary and metastatic OS tumor samples. By applying the CIBERSORT algorithm, they analyzed immune cell infiltration and found that macrophages represent the most prevalent cells, particularly the M0 and M2 subtypes [23]. Similarly, Cao et al., taking advantage of datasets from different databases, such as GEO, Oncomine, and R2, investigated the relationship between the expression of bone morphogenetic protein receptor 2 (BMPR2) and immune tumor-infiltrating cells in OS using again the CIBERSORT and another analytical tool, the Tumor Immune Estimation Resource (TIMER) [24]. They found that BMPR2 expression was significantly higher in OS tissue compared to normal tissue and was correlated with poor prognosis. Interestingly, its expression level showed a significant negative correlation with genes related to CD8^+^ T cells, monocytes, and M2 macrophages, suggesting that low levels of infiltrating CD8^+^ T cells, monocytes, and M2 macrophages in osteosarcoma was significantly associated with poor survival. As for immunohistochemical data, also using bioinformatic methods, the correlation between macrophage infiltration and patient prognosis is still not clear, with some evidence indicating a pro-tumoral role for M2 macrophages, in line with most findings of other tumor types, and others suggestive of an anti-tumor activity, as the work mentioned just above [24].

Interestingly, Deng and colleagues investigated the dynamic change of immune cells before and after neo-adjuvant chemotherapy by GEP and CIBERSORT analysis of matched biopsy and surgical samples from 27 patients. Neo-adjuvant chemotherapy was associated with an increase in CD3^+^ T cells, CD8^+^ T cells, Ki67^+^/CD8^+^ T cells, and PD-L1^+^ immune cells and a decrease in HLA-DR-CD33^+^ MDSCs [25]. This evidence indicates that neo-adjuvant chemotherapy may change the local tumor immune landscape toward a more immune-infiltrated environment, making OS tumors more suitable for immunotherapy. 

Many other works have exploited different bioinformatics strategies to investigate the tumor microenvironment in OS [66,67,68,69], providing new data regarding the different immune cell types that infiltrate OS tumors. However, most of these works rely only on bioinformatics analysis, without providing any experimental confirmatory data, such as protein detection by IHC.

A very elegant recent work that is worth mentioning and reading investigated the landscape of intra-tumoral heterogeneity and the immunosuppressive microenvironment in advanced OS by using single-cell RNA sequencing [22]. The authors presented a comprehensive analysis of the transcriptomic profiling of more than 100,000 single cells from primary osteoblastic and chondroblastic OS lesions, identifying 11 major cell clusters and characterizing the cellular properties of both malignant cells and the TME. Using TME single-cell deconvolution, they identified differences in myeloid cells, lymphocytes, MSCs, and fibroblasts. In the myeloid compartment, they found that monocytes and macrophages account for 70–80% of cells and identified three major subtypes of macrophages in OS lesions, namely, M1-, M2-, and M3-TAMs, being the M3 alveolar FABP4^+^ macrophages mainly present in the lung metastatic lesions. Interestingly, analyzing the T cell compartment, they found that CD8^+^, CD4^+^, Tregs, and NKT cells in OS tumors express at high level the inhibitory immune checkpoint marker TIGIT and demonstrated that blocking TIGIT in vitro enhances the cytotoxic efficacy of CD3 T cells against OS cells, suggesting that the targeting of TIGIT may have potential therapeutic efficacy in the treatment of OS in the future. The authors also investigated in-depth OS-intrinsic characteristics, finding a specific gene expression pattern for lung metastasis or recurrent osteoblastic OS cells compared to the primary osteoblastic OS cells. Similarly, they evaluated the copy number variation (CNV) of the different lesions and found a great amount of genomic CNV pattern shared between the chondroblastic and osteoblastic subclones in the chondroblastic OS lesions, suggesting that a trans-differentiation from malignant chondroblastic cells into malignant osteoblastic cells occurs [22]. 

## 4. Recent Advances and Future Perspectives

The recent advances in comprehensive molecular profiling, including spatial transcriptomics, have increased the understanding of OS confirming its complexity and heterogeneity. Furthermore, the technological progress has been paralleled, in the last two decades, by the creation of well-annotated tissue banks in different countries, including the U.S. and the U.K. and the European Union; considering the rarity of the disease, the set-up of tissue banks has greatly increased the availability of OS samples, thereby facilitating a large number of molecular analysis of pathways and genomic signatures. In the last few years, advances have also been made in OS preclinical models to test the activity of different therapeutic approaches, with the development of a large number of PDX (patient-derived xenograft) models [70]. Interestingly, the molecular alterations identified by molecular profiling of OS samples from patients were mostly recapitulated in the PDX models. This finding clearly indicates that PDX models could be used to test the response to genome-informed treatment [71]. The major disadvantage of PDX tumors is the need of immune-compromised mice for their growth that do not allow the evaluation of the adaptive immune response and limit the possibility of testing immunotherapy-based approaches. An alternative strategy to bypass such an issue is the use of humanized mice to implant the PDX models. 

Taking advantage of the recent technological progress in molecular profiling and drug screening, of the creation of tissue banks and of PDX models, the future in OS management would probably lay in the possibility of running biomarker-based small clinical trials to increase the chance that a specific drug is efficacious in such a heterogeneous patient population.

## 5. Conclusions

Osteosarcomas are very heterogenous tumors, both at the intra- and inter-tumor levels, with no driving mutation identified thus far. These two features render the development of new therapeutic strategies very difficult, and indeed, the standard treatment for OS nowadays is still based on extensive surgical resection and multi-drug chemotherapy. However, in the case of metastatic or recurrent disease, such a therapeutic option is poorly effective and patient prognosis very dismal.

Additionally, OS arises in a the very specialized BM environment in which the different cell types, including immune cells, MSCs, and bone cells, are tightly interconnected and cross-regulate one another, together with OS cells. Therefore, a deeper insight not only into the cellular and molecular characteristics of OS, but also its TME properties and components, will be critical for the development of new therapeutic options in the future. In light of this, the re-programming of the OS microenvironment in order to re-activate the immune system toward an efficient anti-tumor response represents one of the next challenges in OS treatment. Some indications supporting the feasibility of this strategy derive from a few recent works that described the dynamic change of the OS immune landscape before and after neoadjuvant chemotherapy. These works proved that after neo-adjuvant chemotherapy, there is an increased number of TILs that are also paralleled by a decrease in MDSCs. This evidence suggests that, indeed, neo-adjuvant chemotherapy may modify the local tumor immune landscape toward a more immune-infiltrated environment, making cold tumors hot, which, in turn, may be more suitable than immunotherapy approaches.

## Figures and Tables

**Figure 1 cells-10-01668-f001:**
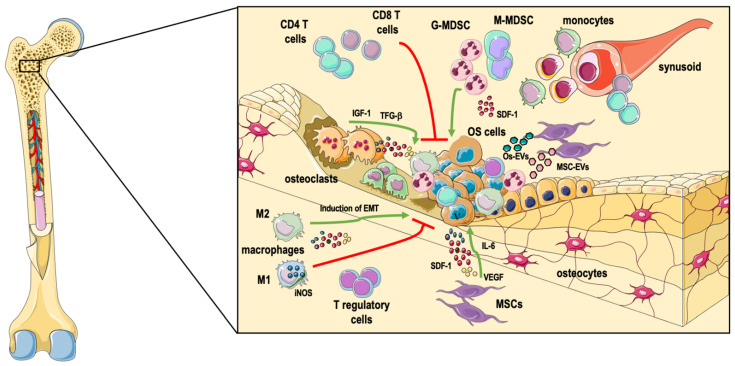
The complex osteosarcoma microenvironment in the bone tissue. Osteosarcoma (OS) develops in the highly specialized bone marrow (BM) environment, a highly dynamic tissue composed of bone cells, immune cells, and stromal and vascular cells, embedded in a mineralized extracellular matrix. The different cell types are reciprocally regulated, with OS cells producing factors that recruit and re-program immune, stromal, and bone cells to their advantage, and vice versa, with immune cells, osteoclasts, and stromal cells releasing pro-tumoral molecules such as VEGF (vascular endothelial growth factor), TGFβ (transforming growth factor beta), and IL-6 (interleukin-6). The cross-talk between OS cells and other BM cells may also occur through the release of extracellular vesicles (EVs). OS cells, osteosarcoma cells; MSCs, mesenchymal stem cells; EMT, epithelial-to-mesenchymal transition; G-MDSC, granulocytic myeloid-derived suppressor cells; M-MDSC, monocytic myeloid-derived suppressor cells. Red line: negative signal; green arrow: positive signal.

**Table 1 cells-10-01668-t001:** Studies on the role of macrophages in osteosarcoma.

Type of Study	Markers and TAM Phenotype	Findings	References
Human samples, GEP and IHC analysis	CD14/HLA-DRα (M1); CD14/CD163 (M2)	Higher CD14^+^ TAMs levels correlated with better OS and metastasis suppression with no clinical relevance between M1/M2 phenotype	[15]
Human samples, IHC analysis	CD163 (M2);	Better overall survival and prolonged metastasis progression-free survival.	[16]
Human samples, IHC analysis	CD68/iNOS/COX-2 (M1); CD163 (M2);	Higher CD68^+^/COX-2 levels in lung metastasis, unchanged CD163^+^ compared to paients with primary tumours while iNOS^+^ was relatively higher.	[17]
Pre-clinical study (THP-1 human cell line)	CD206/CD163 (M2);	BALB/c mice subcutaneously injected with THP-1 cell line promoted migration/invasion of OS cells through MO/M2 TAMs, showing higher levels of ZEB-1 and SNAIL toward an EMT phenotype.	[17]
Human samples, FACS analysis	CD14/CD163 (M2);	Higher CD163^+^ TAMs levels in primary tumour than PB and TILs correlated with lower levels of TIM-3^+^/PD-1^+^ T cells.	[18]
Human samples, IHC analysis	CD68/iNOS (M1);	Higher CD68/iNOS^+^ level in non metastatic patients’ group, with better OS	[19]
Pre-clinical study (U2OS human cell line)	F4/F80/CD163 (M2)	NOD/SCID mice orthotopically injected with OS cells show the recruitment of CD163^+^ M2 TAMs subtype and a higher tumour growth.	[20]
Pre-clinical study (K7M2 mouse cell line)	CD163/CD209/MRC1/CCR2/F4/80 (M2);	BALB/c mice orthotopically co-injected with OS with or w/out RAW264.7 cells treated with ATRA show reduced M2-polarization through suppressing colony/sarcosphere formation and tumour growth.	[21]
Human samples, single cell RNA seq analysis	CD163/MRC1/MS4A4/MAF (M2); FABP4^+^ (M3);	M1-, M2- and M3 TAMs are the 80% of the major macrophages cell subtypes in the myeloid compartment of OS lesions, while the FABP4^+^ M3 levels are directly correlated with the metastatic spread through the lung.	[22]
Human samples, CIBERSORT algorithm	Defined by CIBERSORT	M0 and M2 are the most relevant cell subtypes in patients’ metastatic samples.	[23]
Human samples, CIBERSORT and TIMER analysis	Defined by CIBERSORT	Lower levels of M2 TAMs directly associated with patients’ poor survival.	[24]
Human samples, GEP and TIMER analysis	Defined by CIBERSORT	Higher M1/M2 TAMs levels in the infiltrating microenvironment showed an improved overall survival of patients; better OS	[25]

**Footnotes**. TAMs: Tumor associated macrophages; PB: peripheral blood; TILs: tumor infiltrating T cells; ATRA: all-trans retinoic acid; EMT: epithelial to mesenchymal transition; GEP: gene expression profile; IHC: immunohistochemistry; FACS: fluorescent-activated cell sorting.

## Data Availability

Not applicable.

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
