# Peer review of "The Immune Landscape of Osteosarcoma: Implications for Prognosis and Treatment Response"

_cells, 2021, doi:10.3390/cells10071668_

Round 1

Reviewer 1 Report

This paper is well written and interesting. It presents a comprehensive review of immune cells, hematopoietic cells and mesenchymal stem cells that exert a role in osteosarcoma (OS) progression and metastatic diffusion. The authors methodically summarize important papers in this area. In general, this work could be made more exciting by discussing a number of issues such as: 10 what progresses can we reasonably expect in this field in the near and midterm feature? 2) What interesting techniques are under development that may lead to novel results soon? 3) What are the limitations of the work conducted so far? Do we have adequate animal models?  Should the field attempt to develop better model? 4) Is there a role for the gut microbiome? 5) Why is OS more prevalent in the first 2 decades of life, a time when the skeletal grows rapidly?

I would suggest providing more information about specific T cell subsets including gamma/delta T cells (which are modulated by bisphosphonates), Th17 cells and Tregs.  Does T cell exhaustion play a role?

The discussion about the role of osteoclasts is very interesting. Additional information should be provided about the role, if any, of antiresorptive agents such as Denosumab and long-acting bisphosphonates.

Author Response

Reviewer 1:

This paper is well written and interesting. It presents a comprehensive review of immune cells, hematopoietic cells and mesenchymal stem cells that exert a role in osteosarcoma (OS) progression and metastatic diffusion. The authors methodically summarize important papers in this area. In general, this work could be made more exciting by discussing a number of issues such as: 1) what progresses can we reasonably expect in this field in the near and midterm feature? 2) What interesting techniques are under development that may lead to novel results soon? 3) What are the limitations of the work conducted so far? Do we have adequate animal models?  Should the field attempt to develop better model? 4) Is there a role for the gut microbiome? 5) Why is OS more prevalent in the first 2 decades of life, a time when the skeletal grows rapidly?

I would suggest providing more information about specific T cell subsets including gamma/delta T cells (which are modulated by bisphosphonates), Th17 cells and Tregs.  Does T cell exhaustion play a role?

The discussion about the role of osteoclasts is very interesting. Additional information should be provided about the role, if any, of antiresorptive agents such as Denosumab and long-acting bisphosphonates.

R: We thank the Reviewer for his/her comments and suggestions that we have tried to address at our best.

In general, this work could be made more exciting by discussing a number of issues such as: 1) what progresses can we reasonably expect in this field in the near and midterm feature? 2) What interesting techniques are under development that may lead to novel results soon? 3) What are the limitations of the work conducted so far? Do we have adequate animal models?  Should the field attempt to develop better model? 4) Is there a role for the gut microbiome? 5) Why is OS more prevalent in the first 2 decades of life, a time when the skeletal grows rapidly?

R: We agree with the reviewer that discussing/speculating on the points he/she raised would be very interesting, but we feel that these issues are out of the scope of the review that is focused on the immune microenvironment and moreover, for a matter of space, they could not be discussed in details in this context. We have, nevertheless, briefly touched each of his/her points adding a NEW SECTION (Recent advances and future perspectives) before the Conclusion of the review. We did not introduce gut microbiome since we did not find any report demonstrating a role for gut microbiome in osteosarcoma progression and response to therapy. However, as it has been clearly shown for other tumor types, such as melanoma, the gut microbiome could be relevant especially in terms of response to chemo and immunotherapy.

We have highlighted the changes in the text in yellow for an easier identification.

I would suggest providing more information about specific T cell subsets including gamma/delta T cells (which are modulated by bisphosphonates), Th17 cells and Tregs.  Does T cell exhaustion play a role?

R: As requested by the reviewer we expanded the section on the role of other T cell subsets, discussing the pieces of evidence for a potential activity of T regulatory cells, Th17 and gamma/delta T cells. See section 2.1.3. of the review “T lymphocytes and other immune cell types”.

The discussion about the role of osteoclasts is very interesting. Additional information should be provided about the role, if any, of antiresorptive agents such as Denosumab and long-acting bisphosphonates.

R: As suggested by the reviewer, we expanded the section 2.3 on osteoclasts, adding information on Denosumab and bisphosphonates.

Reviewer 2 Report

The review by Cascini and Chiodoni gives a nice general overview of a cross-talk between bone microenvironment and osteosarcoma. The manuscript  is well written with an intuitive figure. This review mainly covers the roles of immune cells in the tumor microenvironment of osteosarcoma, implication for disease prognosis, and treatment response. However, some point needs to be corrected:

  • The manuscript would benefit from a through revision of grammar.
  • In Figure 1, there is no indication of OS cells.
  • In line 102, “who and” should be removed.
  • In line 144, font size is incorrect.
  • Table 1 needs to fix the alignment.
  • In line 156, describe what metformin is.
  • In 2.2, “mesenchymal stromal cells” should be changed to “mesenchymal stem cells” or vice versa.
  • In line 323, “100.000” should be changed to “100,000”.

Author Response

Reviewer 2:

The review by Cascini and Chiodoni gives a nice general overview of a cross-talk between bone microenvironment and osteosarcoma. The manuscript is well written with an intuitive figure. This review mainly covers the roles of immune cells in the tumor microenvironment of osteosarcoma, implication for disease prognosis, and treatment response. However, some point needs to be corrected:

  • The manuscript would benefit from a through revision of grammar.
  • In Figure 1, there is no indication of OS cells.
  • In line 102, “who and” should be removed.
  • In line 144, font size is incorrect.
  • Table 1 needs to fix the alignment.
  • In line 156, describe what metformin is.
  • In 2.2, “mesenchymal stromal cells” should be changed to “mesenchymal stem cells” or vice versa.
  • In line 323, “100.000” should be changed to “100,000”.

R: We thank the Reviewer for his/her comments and suggestions that we have addressed as detailed below. We modified the text and figure accordingly and highlighted in yellow the changes.

  • The manuscript would benefit from a through revision of grammar.
  • R: We have the manuscript revised by an English native speaking colleague.
  • In Figure 1, there is no indication of OS cells.
  • R: We modified the figure adding the text “OS cells”.
  • In line 102, “who and” should be removed.
  • R: We believe the sentence is correct since the gene expression included BOTH patients who has developed metastases and patients who did not develop metastases within 5 years from the diagnosis.
  • In line 144, font size is incorrect.
  • R: Font size is now 10 pts as the wholw text
  • Table 1 needs to fix the alignment.
  • R: We have fixed the Table (however the wrong alignment was not due to our original file but it depended on the format edited by the journal, therefore we are not sure it will remain correct).
  • In line 156, describe what metformin is.
  • R: We have modified the sentence describing what is metformin.
  • In 2.2, “mesenchymal stromal cells” should be changed to “mesenchymal stem cells” or vice versa. R:
  • We have changed “mesenchymal stromal cells” with “mesenchymal stem cells”.
  • In line 323, “100.000” should be changed to “100,000”.
  • R: We have changed the number accordingly.

Reviewer 3 Report

The authors summarize the current knowledge about the tumor microenvironment in osteosarcoma, focusing on immune cells including macrophages, other myeloid cells, lymphocytes, mesenchymal stromal cells, and osteoclasts. Also, they discussed their roles and implication for disease prognosis. The manuscript is well written and substantially overviewed the research field.

I have some suggestions to improve the manuscript.

#1. Each section needs to be updated by referring to recent findings. The authors referred to a recent paper (Zhou Y, et al. Nat Commun 2020, 11) as Ref #56 at the end of the main text. The paper has provided novel findings in this research field, and the content is closely related to the topic of this review. The novel findings should be described in each section of this review, not just citing briefly at the end of the manuscript.

#2. In section 2.3. osteoclasts, the authors described that osteoclast-ablative therapies could affect OS tumor growth. But, the authors did not discuss the mode of action of the OC-ablative therapies. In Ref #46, they used RANK-Fc, which could affect osteoblasts and OS themselves, whereas, in Ref #47, they used a bisphosphonate, which directly affects osteoclasts. The MOA of the drugs should be discussed.

#3. In section 2.3. osteoclasts, recent papers regarding the effect of Denosumab on OS can be added.

Minor point

#4. TGFb should be replaced with TGFβ in the text of the Figure and the figure legend.

Author Response

Reviewer 3:

The authors summarize the current knowledge about the tumor microenvironment in osteosarcoma, focusing on immune cells including macrophages, other myeloid cells, lymphocytes, mesenchymal stromal cells, and osteoclasts. Also, they discussed their roles and implication for disease prognosis. The manuscript is well written and substantially overviewed the research field.

 I have some suggestions to improve the manuscript.

 #1. Each section needs to be updated by referring to recent findings. The authors referred to a recent paper (Zhou Y, et al. Nat Commun 2020, 11) as Ref #56 at the end of the main text. The paper has provided novel findings in this research field, and the content is closely related to the topic of this review. The novel findings should be described in each section of this review, not just citing briefly at the end of the manuscript.

 #2. In section 2.3. osteoclasts, the authors described that osteoclast-ablative therapies could affect OS tumor growth. But, the authors did not discuss the mode of action of the OC-ablative therapies. In Ref #46, they used RANK-Fc, which could affect osteoblasts and OS themselves, whereas, in Ref #47, they used a bisphosphonate, which directly affects osteoclasts. The MOA of the drugs should be discussed.

 #3. In section 2.3. osteoclasts, recent papers regarding the effect of Denosumab on OS can be added.

 Minor point #4. TGFb should be replaced with TGFβ in the text of the Figure and the figure legend.

R: We thank the Reviewer for his/her comments and suggestions that we have addressed as detailed below. We modified the text and figure accordingly and highlighted in yellow the changes.

 #1. Each section needs to be updated by referring to recent findings. The authors referred to a recent paper (Zhou Y, et al. Nat Commun 2020, 11) as Ref #56 at the end of the main text. The paper has provided novel findings in this research field, and the content is closely related to the topic of this review. The novel findings should be described in each section of this review, not just citing briefly at the end of the manuscript.

R: We have updated some part of the review with more recent papers, however it should be mentioned, that for some of the issues discussed we could not find very recent publications.

Regarding the paper mentioned by the reviewer, we have extended its discussion in the sections we believed to be the most appropriate for the specific paper (adding a citation also in the 2.3 section on osteoclasts). We agree with the reviewer that the paper is really interesting and relevant for OS and we believe we have underlined its relevant with our words “A very elegant recent work that it’s worth to be mentioned and read…”. However, for a matter of “equity” and balance with other papers, we did not refer to it in each section. We believe that our underlining of the relevance of this work will encourage the readers to look up the whole paper by their own.

 #2. In section 2.3. osteoclasts, the authors described that osteoclast-ablative therapies could affect OS tumor growth. But, the authors did not discuss the mode of action of the OC-ablative therapies. In Ref #46, they used RANK-Fc, which could affect osteoblasts and OS themselves, whereas, in Ref #47, they used a bisphosphonate, which directly affects osteoclasts. The MOA of the drugs should be discussed.

R: As suggested by the reviewer, we expanded the section 2.3 on osteoclasts, adding information on Denosumab and bisphosphonates and discussing their mechanisms of action.

 #3. In section 2.3. osteoclasts, recent papers regarding the effect of Denosumab on OS can be added.

R: As mentioned above we expanded the section 2.3.  However, there are no recent papers on Denosumab on OS, being Denosumab mainly used for other bone primary tumors (giant cell tumors) or for bone secondary tumors. We added a very elegant recent paper on the potential mechanisms of RANK-L role in OS aggressiveness and on the blockade of RANK-L in OS genetically-modified mouse models (Chen, Sci Transl Med. 2015 PMID: 26659571).

#4. TGFb should be replaced with TGFβ in the text of the Figure and the figure legend.

R: We have replaced TGFb with TGFβ in the text and the figure legend as requested, sorry for the mistake.

Round 2

Reviewer 1 Report

I am satisfied with the revisions

Reviewer 3 Report

This second version of the paper has been greatly improved by adding a detailed description.

I have no additional comment.